# The 8th Wonder of the Cancer World: Esophageal Cancer and Inflammation

**DOI:** 10.3390/diseases10030044

**Published:** 2022-07-07

**Authors:** Harleen Kaur Chela, Karthik Gangu, Hamza Ertugrul, Alhareth Al Juboori, Ebubekir Daglilar, Veysel Tahan

**Affiliations:** 1Department of Gastroenterology, University of Missouri, Columbia, MO 65201, USA; he19md@gmail.com (H.E.); alhareth.aljuboori@gmail.com (A.A.J.); edaglilar@health.missouri.edu (E.D.); tahanv@health.missouri.edu (V.T.); 2Department of Medicine, Division of Hospital Medicine, University of Missouri, Columbia, MO 65201, USA; karthik.gangu@health.missouri.edu

**Keywords:** cancer, pathogenesis, inflammation, esophagus, pathology

## Abstract

Esophageal cancer is a devastating malignancy which can be detected at an early stage but is more often diagnosed as an advanced process. It affects both men and women and inflicts the young and the elderly. There are multiple underlying factors involved in the pathogenesis of this cancer including inflammation. The interplay of these factors promotes inflammation through various mechanisms including the recruitment of pro-inflammatory cells, mediators such as cytokines, reactive oxygen species, and interleukins, among others. The presentation can vary widely with one of the most notable symptoms being dysphagia. Diagnosis is based on clinical symptomatology, imaging and endoscopy with biopsy. Once the diagnosis has been established, treatment and prognosis are based on the stage of the disease. This review outlines esophageal cancer and its link to inflammation in relation to pathogenesis, along with clinical features, diagnosis and treatment.

## 1. Introduction

Esophageal cancer is rapidly increasing across the world [1]. It is the sixth most common cause of death from cancer and is the eighth most common cancer worldwide [2]. There are two main types of cancer of the esophagus based on the cell of origin: squamous cell carcinoma and adenocarcinoma. Other uncommon forms of esophageal malignancy include melanomas, sarcomas, lymphomas and carcinoid tumors [3]. Squamous cell carcinoma originates from squamous epithelial cells of the inner lining of the esophagus and is located in the proximal to middle portions of the esophagus [4]. Adenocarcinoma originates from the glandular/secretory cells lining the distal esophagus [4]. Based on the location, the incidence and distribution of the type of esophageal cancer varies [5]. Typically, esophageal squamous cell carcinoma is thought primarily be a disease of the developing world, and adenocarcinoma a disease of the developed world. The prognosis is typically poor and the five-year survival rate ranges from 5–15% [2]. Esophageal squamous cell carcinoma (ESCC) is the predominant histological variant globally and is more common in developing countries in the eastern parts of the world, such as Asia and Africa [2]. An entity known as the Asian Esophageal Cancer Belt has a high incidence of esophageal squamous cell carcinoma and includes countries such as Iran, parts of China, Turkey, and Kazakhstan [2]. However, in North America and Europe, esophageal adenocarcinoma (EAC) is the main type of esophageal cancer that is encountered and is related to the development of Barrett’s esophagus among other factors [3,6]. There has been a drastic change in incidence with a decline in SCC and increase in EAC [7,8]. There are differences in the distribution of the types of esophageal cancer and the underlying predisposing factors. There are multiple factors involved in the pathogenesis (Table 1).

Inflammation, especially chronic, is involved in the pathogenesis of many malignancies, including esophageal cancer. Due to long standing exposure to inflammatory triggers, an inflammatory milieu is created with the presence of pro-inflammatory cells that generate cytokines. Inflammatory cells such as neutrophils, natural killer cells, monocytes, and macrophages are recruited to areas of ongoing injury [9]. These inflammatory cells, once activated, produce reactive oxygen species (ROS), which promotes carcinogenesis by stimulation of oncogenes and the suppression of tumor suppressor genes along with mutations in the DNA [9,10]. Figure 1 depicts some of the factors involved in the pathogenesis.

## 2. Pathogenesis

### 2.1. Etiological Factors

#### 2.1.1. Smoking

Smoking is a risk factor for both ESCC and adenocarcinoma of the esophagus [2,5]. Smoking predisposes to the development of Barrett’s and also increases the risk for adenocarcinoma [2]. Smoking duration and amount is also correlated to the overall risk [11]. The magnitude of risk is also higher for squamous cell carcinoma as compared to adenocarcinoma [11]. The risk for male smokers is noted to be increased as compared to female smokers [12]. Smoking generates a pro-inflammatory response with production of several inflammatory cytokines and molecules. It promotes an inflammatory state by inducing the generation of cytokines such as interleukin (IL)-1, IL-6 and IL-8 and tumor necrosis factor-α (TNF-α) [13]. It reduces the levels of IL-10, which is an anti-inflammatory medicator [13]. The NF-kB (Nuclear factor kappa B) pathway is one of the main factors underlying the stimulation of inflammatory cells secondary to smoking [14]. Tobacco smoke is also a source of free radicals such as ROS and reactive nitrogen species (RNS) that lead to oxidative stress [15]. These include hydrogen peroxide, hydroxyl radicals, superoxide, and nitric oxide [15]. These reactive oxygen species and reactive nitrogen species lead to the activation of the NF-kB pathway and the creation of a pro-inflammatory state [16]. These reactive nitrogen and oxygen species cause oxidative stress and damage to lipids, proteins and nucleic acids, which not only lead to inflammation but contribute to carcinogenesis [17]. The peroxidation of lipids can cause the formation of products that cause damage to DNA and proteins [18].

#### 2.1.2. Alcohol

Alcohol is metabolized by the enzyme alcohol dehydrogenase, with ethanol being converted to acetaldehyde [9]. Acetaldehyde exerts a carcinogenic effect by forming DNA adducts and alters the genes to lead to a mutation [9]. The consumption of alcohol is clearly linked to the development of ESCC but not as strongly to adenocarcinoma [11]. A prospective cohort study conducted in the Netherlands did not find a significant link between adenocarcinoma and alcohol use [19]. However with regard to ESCC, the risk can be up to three to five times greater in those who have a history of alcohol consumption [19,20]. In a large meta-analysis of case control and cohort studies conducted by Islami et al., it was seen that moderate and high consumption of alcohol was linked to a higher risk of ESCC in Asian and non-Asian countries [21]. Light consumption of alcohol was only noted to be connected to a higher risk for ESCC in Asian countries [21]. However, a dose-dependent association was seen between alcohol intake and risk of ESCC in Asian and non-Asian countries [21]. The type of alcoholic beverages and association with esophageal malignancy is not as clearly established [11].

Alcohol can lead to inflammation in the gastrointestinal tract along with dysbiosis, bacterial overgrowth and intestinal hyperpermeability [22]. Alcohol is metabolized predominantly in the hepatocytes, with conversion of alcohol to acetaldehyde via the enzyme alcohol dehydrogenase [22]. The acetaldehyde, which is harmful, is then metabolized to acetate via the enzyme acetaldehyde dehydrogenase [22]. The oxidative metabolism of alcohol also occurs in the intestines and bacteria generates acetaldehyde in the gut [23]. In cases of chronic alcohol consumption, there is a nonoxidative metabolism of alcohol in the intestines that involves phospholipids and free fatty acids and can have harmful effects [24]. The microsomal ethanol–oxidizing system is another form of metabolism in which alcohol is subjected to and leads to the generation of free radicals with resultant injury to cells [22]. By increasing the permeability of the wall of the intestines, alcohol permits bacteria to translocate, and due to the release of endotoxins cause local and systemic inflammation with the release of cytokines [22]. Alcohol also promotes an inflammatory state by decreasing immunity and lowering the activity of the Paneth cells [22].

#### 2.1.3. Diet

Diet can play a pivotal role in the predisposition of esophageal malignancy. A diet high in meat and fast foods akin to the Western diet increases the risk for Barrett’s esophagus and subsequent adenocarcinoma [25]. Whereas a diet that is abundant in fruits, vegetables, nonfried fish is linked to a decreased risk for cancer of the esophagus, as all of these contain antioxidants [25]. A diet that is high in omega-3-fatty-acids, total fiber, polyunsaturated fats, and fiber is linked to a decreased risk for Barrett’s esophagus [26]. The intake of vitamin D, beta-carotene, and vitamin C also has protective effects against Barrett’s esophagus [27]. Consumption of meat can lead to the development of esophageal cancer through the production of mutagenic heterocyclic amines and polycyclic aromatic hydrocarbons when it is prepared at high temperatures [28]. Meat is a source of iron and a source of nitrates and nitrites when processed, and this can lead to the generation of N-nitroso compounds, which have carcinogenic properties [28]. Though ESCC shares some similar underlying risks, there are other dietary factors that are more pertinent to the development of ESCC.

In developing countries across the world such as in Northern China and Northeastern Iran, the dietary factors leading to ESCC have been well studied. The Linxian area in Northern China has a high incidence and mortality from ESCC [29]. A diet lacking in fruits and vegetables along with a deficiency of micronutrients such as carotenoids, riboflavin and vitamins A, C and E is associated with a risk for ESCC [29]. Diet-derived carcinogens such as nitrosamines and polycyclic aromatic hydrocarbons also predispose to ESCC [29]. Preformed nitrosamines and endogenously produced nitrosamines both play a role in carcinogenesis [29]. Nitrosamines are generated from nitrites (formed by the oxidation of nitrogenous elements in the water) and secondary amines (can be found in substances such as moldy corn) [29]. A nitrosation reaction of moldy corn can produce carcinogens such as methylbenzylnitrosamine, diethylnitrosamine and N-1-methylacetonyl-N-3-methylbutylnitrosamine [30]. Polycyclic aromatic hydrocarbons are formed due to the burning of wood and coal and have harmful effects as well [30]. The deficiency of micronutrients such as selenium and zinc may also be linked to esophageal cancer [31].

The ingestion of pickled vegetables has also been shown in some studies to be a risk factor for ESCC [30]. The proliferation of yeasts and fungi in the pickled vegetables can lead to formation of harmful substances such as mycotoxins, Roussin red methyl ester and N-nitrosamines [30,32,33]. Thermal damage to the lining of the esophagus can occur from the ingestion of hot food and beverages and increase the risk for ESCC [34]. The chewing of betel quid, which is common in areas like India and Taiwan, can cause irritation of the esophageal mucosa and lead to ESCC, and risk increases in conjunction with the use of tobacco [35]. The Mediterranean diet, which is increasing in popularity, had been found to have some beneficial effects with regard to esophageal cancer. A Mediterranean diet is plant based and is rich in whole grains, vegetables, and fruits, and the predominant fat is olive oil [36]. A study conducted in the Netherlands revealed that consuming a Mediterranean diet was related to a decreased risk for both ESCC and EAC [36]. This diet has high levels of anti-oxidants and can decrease oxidative damage of the DNA and reduce inflammation [37]. The presence of dietary fiber as well in this diet can be protective against the mutagenic properties of N-nitroso compounds by being a nitrite scavenger [37]. As the Mediterranean diet is low in meat, the harmful effects that occur due to meat as described above are much less [28].

Maté is made from a herb called Ilex paraguayensis, and when hot maté is consumed, especially in large amounts, it is linked to the development of esophageal cancer [31,38]. The recurrent thermal injury which occurs due to ingestion of hot maté and the polycyclic aromatic hydrocarbons that it contains lead to carcinogenesis [38]. Carbonated soft drinks have been speculated to have an association with EAC due to increased reflux secondary to gaseous distention of the stomach and their acidic nature [31]. However, other studies have not shown any correlation and hence the findings are inconclusive [31].

#### 2.1.4. Gastro-Esophageal Reflux Disease (GERD)

The presence of uncontrolled and untreated reflux is an important factor in the pathogenesis of Barrett’s esophagus and the development of esophageal cancer [12]. Longstanding GERD induces acid related damage to the esophageal lining leading to chronic inflammation in the esophagus (Figure 2) and the eventual metaplastic change known as Barrett’s esophagus (Figure 3). The intestinal metaplasia with columnar cells and goblet cells is believed to be more resilient towards the deleterious effects of chronic acid exposure [39]. Continuous exposure to inflammation causes the production of substances such as cytokines, chemokines, and reactive oxygen species that can induce harmful effects [9]. They can ultimately lead to augmentation of cell growth, stimulate invasion and promote the growth of blood vessels [9]. Inflammatory cells may also produce mediators that can inhibit immune functions, and this also promotes carcinogenesis [40]. This progresses over time to dysplastic and neoplastic changes, as Barrett’s esophagus is a well-known risk factor for adenocarcinoma of the esophagus. GERD is not known to be a risk factor for ESCC.

#### 2.1.5. Obesity

Obesity has been shown to have an association with adenocarcinoma with a higher BMI and abdominal obesity having a linkage to EAC [41]. A BMI greater than 25 kg/m^2^ raises the risk for adenocarcinoma in males and females, and the higher the BMI the greater the risk [41]. Obesity raises the intra-abdominal pressure and this worsens reflux and thereby predisposes the development of Barrett’s esophagus and the subsequent risk for EAC [42]. Inversely, an increased BMI is associated with a decreased risk for ESCC [12]. Obesity is also associated with a chronic state of low inflammation that involves adipocytes [43]. It leads to elevated levels of C-reactive protein and adipocyte induced activation of pathways of inflammation [43]. Cytokines that promote inflammation are produced by the adipocytes as well as immunological cells (such as T lymphocytes and macrophages) [44]. T lymphocytes produce interferon-gamma (INF-γ), IL-1, and IL-17 and the macrophages produce IL-12 and TNF-α [45]. Other cytokines that may play a causative role in carcinogenesis in the state of obesity include insulin-like growth factor-1 (IGF-1), transforming growth factor-beta (TGF-β), and vascular endothelium growth factor [46]. These factors are linked to cell proliferation and the promotion of angiogenesis and hence the possible causal link to carcinogenesis [46].

#### 2.1.6. Infections

There are some infectious agents that have been speculated to be implicated in the pathogenesis of esophageal cancer. Bacteria such as *Helicobacter pylori* and viral agents such as the *Human papillomavirus types 16 and 18* may contribute to the development of esophageal cancer. *Helicobacter pylori* is a possible factor in ESCC [47], but results are inconclusive, as other studies show no correlation. For EAC, *H. pylori* shows a protective effect according to some studies and was found to have a decreased risk for EAC [48,49]. *H. pylori* lessens the generation of acid in the stomach and leads to decreased reflux of gastric acid, thereby mitigating the effect for Barrett’s and EAC [50]. *H. pylori* may also reduce the generation of ghrelin, which enhances appetite and is a hormone formed in the stomach [51]. By decreasing appetite, a lower level of ghrelin may decrease obesity and hence lower the risk for EAC [52]. Due to use of antibiotics and improvements in sanitation, there is a decreased colonization of *H. pylori* in western countries, and this could be contributing to the increase in EAC [31,53]. The human papilloma virus (HPV) has oncogenic variants such as HPV 16 and 18, and though the findings are controversial, there are some studies that implicate these variants in the pathogenesis of esophageal cancer [54]. Epstein-Barr virus (EBV) infects the B lymphocytes and has known oncogenic potential, but with regards to esophageal cancer, the available data is conflicting, with some studies showing an association but others showing none [55].

#### 2.1.7. Anatomical Factors

Anatomical changes such as hiatal hernia, achalasia, and gastric atrophy can all predispose to esophageal cancer. A hiatal hernia, especially one that is large, can lead to a higher risk for EAC, as it increases the reflux of gastric acid and can cause the development of Barrett’s esophagus [56,57]. This is not considered to be a risk factor for ESCC. Gastric atrophy leads to increased risk for ESCC as noted by some studies that show that decreased levels of serum pepsinogens are linked to a higher risk for ESCC [58]. When there is gastric atrophy there is decreased production of gastric acid, and bacteria can grow in the stomach and produce carcinogenic compounds such as acetaldehyde and nitrosamines [31,59]. Achalasia is a disorder of esophageal dysmotility that is characterized by lack of peristalsis in the distal esophagus along with the absence of lower esophageal sphincter relaxation [31]. Due to the stagnation of food and subsequent fermentation, it leads to inflammation and is linked to a risk for esophageal malignancy [60]. A large study conducted in Sweden showed that achalasia increased the risk for both esophageal adenocarcinoma and squamous cell carcinoma [61].

#### 2.1.8. Pre-Malignant Esophageal Disorders

Certain conditions of the esophagus are considered premalignant due to their predisposition to esophageal cancer. Tylosis, also called hyperkeratosis palmaris et plantaris, is an autosomal dominant disorder that is linked to a very high risk for developing esophageal cancer [62]. Genetic mutations involving RHBDF2 located on 17q25.1 are believed to occur [62]. Esophageal involvement in the form of small, white colored polypoid areas are seen in the esophagus, along with oral leukokeratosis and cutaneous manifestations [62]. The ingestion of caustic substances such as lye have been shown to cause not only the formation of strictures but also the development of esophageal cancer [63]. Plummer-Vinson syndrome is another pre-cancerous condition involving the esophagus that is characterized by the triad of iron deficiency anemia, dysphagia and esophageal web [64]. It is seen more frequently in women of middle age and has a significant association with the development of squamous cell carcinoma [64].

#### 2.1.9. Genetic Factors

Changes at the genome level are known to play a part in the pathogenesis of esophageal cancer. Genes that regulate the cell cycle are mutated (such as CDKN2A, NFE2L2, RB1, CHEK1, CHEK2) or are over-expressed (such as CCND1, CDK4/CDK6, MDM2) in cases of ESCC [65]. Mutations in genes that are involved in cell differentiation (NOTCH1, NOTCH3) also occur in ESCC [65]. Over expression of epidermal growth factor receptor is seen in cases of ESCC and is linked to worse prognosis [66]. Mutations in receptor tyrosine kinase and RAS signaling pathways also occurs in ESCC [65]. Epigenetic changes such as histone modification, DNA methylation, and loss of genome imprinting are also implicated in the development of ESCC [67]. In EAC the expression of B-cell translocation gene 3 (encodes protein that modulates progression of cell cycle) [68] and the level of which has been linked to lymph node metastasis as well as tumor staging [68,69]. Increased expression of vascular endothelial growth factor (VEGF)-C has been seen in adenocarcinoma along with high levels of cyclin E (encoded by cyclin E1 gene), which plays a role in tumor progression [12,70].

## 3. Clinical Features

Esophageal cancer can have varying degrees of presentation, as early in the disease process it may be silent and symptomatic [3]. However, unfortunately many patients present when the disease progresses; it manifests as progressive dysphagia to solids then to liquids, odynophagia, and the unintentional loss of weight [3]. Refractory heartburn, the new onset of dyspepsia, atypical chest pain as well as signs of gastrointestinal blood loss can occur (occult or overt) [3]. Microcytic anemia can occur as a result of blood loss. Cervical lymphadenopathy as well as hoarseness of the voice (secondary to involvement of the recurrent laryngeal nerve) can occur [71]. Aside from that, the patient may have halitosis, and involvement of the diaphragm can cause hiccups to occur [72].

## 4. Diagnosis

The diagnosis of esophageal cancer is based on clinical suspicion which stems from a thorough history and physical examination. Further evaluation is then performed with imaging as well as endoscopy (Table 2). Endoscopy with biopsy yielding the tissue diagnosis is the definitive form of diagnosis. Once diagnosis is established then imaging modalities will typically be used for staging of the malignancy. The TNM classification system is used for staging of esophageal cancer where the ‘T’ portion evaluates the depth of invasion of the tumor into the wall of the esophagus and beyond. ‘N’ evaluates the presence or absence of nodal metastasis and the extent, whereas the ‘M’ indicates the presence or absence of distal metastasis. Lesions that invade the lamina propria or muscularis mucosae are known as T1a and have less than 10% probability of spreading to the lymph nodes [73]. The T1b lesions involve the submucosa and is linked to a 30% chance of nodal metastasis [73].

### 4.1. Barium Esophagogram

On esophagogram, esophageal cancer can appear as an irregular ulceration of the mucosa, an irregular appearing stricture or as polypoidal appearing filling defects within the lumen of the esophagus [74]. When the tumor burrows through the wall of the esophagus it can lead to extravasation of contrast into the mediastinum in cases of esophageal perforation and into the trachea in cases of trachea-esophageal fistula [75].

### 4.2. Esophagogastroduodenoscopy

Upper endoscopy is considered as the gold standard for establishing the diagnosis of esophageal cancer [73]. During endoscopy, abnormalities such as a suspicious nodule or mucosal abnormality or even a frank mass may be visible [73]. The suspicious area can be biopsied and a histological tissue can be established hence confirming the diagnosis. Ideally, a pathologist who specializes in gastrointestinal disease should be available to review the tissue samples obtained during endoscopy. Especially in cases of Barrett’s esophagus with any dysplasia, two pathologists that have training in gastroenterology are required to review and confirm the diagnosis [76]. However, there are limitations with endoscopy, as it cannot evaluate for locoregional lymph node metastasis or distant metastasis [73].

### 4.3. Multidetector Computed Tomography (MDCT)

Imaging with an MDCT scan can be used to identify the regional and distal spread of malignancy. It can identify T3 disease, which is characterized by involvement of the paraesophageal tissue such as the adventitia without spread to nearby structures [77]. T4 disease is diagnosed by the presence of metastases to adjacent structures including the surrounding organs, vasculature, and bony structures [77]. Invasion of the periesophageal fat is consistent with T3 disease, and loss of the planes of fat between the surrounding structures and the tumor itself is seen with T4 disease [78]. The invasion of blood vessels such as the aorta can be identified and the invasion of the trachea can be diagnosed by indentation of the trachea and by the formation of a trachea-esophageal fistula [75,78]. It can identify local lymph node involvement as well [73]. Aside from radiation, another disadvantage is that MDCT cannot stage T1/T2 lesions [75].

### 4.4. Positron Emission Tomography-Computed Tomography (PET-CT) Scan

As compared to MDCT, a PET-CT scan has a greater sensitivity for identifying esophageal cancer, although it has limitations in T staging except for detecting the infiltration of mediastinal organs [79]. It can be used for the identification of an occult primary tumor in cases of widespread metastasis but is not able to adequately delineate between T1, T2 and T3 lesions [75].

### 4.5. Endoscopic Ultrasound (EUS)

Endoscopic ultrasonography is a diagnostic modality that is utilized for the locoregional staging of disease (T and N staging) but cannot clearly delineate between T1a or T1b lesions at times [73]. Depending on the extent of infiltration of the esophageal wall, EUS can distinguish between the T1/T2 and T3/T4 stages, and this carries implications for surgical management [75]. The accuracy of staging is enhanced with the level of invasion [75]. Lymph nodes on EUS can appear as round, well demarcated hypoechoic structures that are homogenous [73]. Fine needle aspiration of these lymph nodes can identify the presence of tumor metastasis [80]. One of the drawbacks of this technique is that it is operator dependent, inadequate assessment of stenotic growths and the risks associated with endoscopy itself.

## 5. Treatment

The treatment of esophageal cancer involves a multi-disciplinary approach with involvement of the gastroenterology, hematology/oncology, radiation oncology, pathology, surgical oncology, nutrition and palliative care teams. Once the diagnosis is established and staging is completed, then clinical management can be pursued with the patient’s wishes kept in mind. Apart from staging, the location, the histology, age of the patient and their comorbidities, the approach to treatment is determined [81,82]. The treatment may be endoscopic, surgical, neoadjuvant therapy with surgery, may involve chemoradiation, or be palliative in nature.

### 5.1. Endoscopic Therapy

Endoscopic treatment can be pursued with a curative intent or with a palliative intent. Prior to embarking on endoscopic treatment, careful decision-making is needed regarding patient selection based on stage, comorbidities, risks and benefits involved. In early disease with high grade dysplasia (Tis), mucosal (T1a) or submucosal (T1b) invasion, endoscopic management can be pursued as the primary modality (Figure 4). Endoscopic treatment in the form of endoscopic mucosal resection (EMR) or endoscopic submucosal dissection (ESD) can be successfully achieved [83,84]. However, the involvement of surgical oncology at the time of diagnosis is important along with hematology/oncology teams. As based on the results of the surgical specimen, subsequent surgery or perhaps chemoradiation may be necessary in those who are not suitable [82]. Factors that may influence the need for surgery include any lymphovascular involvement, deeper submucosal infiltration, and poor differentiation [85,86]. Differentiation of T1a from T1b disease is crucial given the higher risk for lymph node metastasis in T1b disease. T1a disease is limited to the mucosa and is further subdivided into M1 (intraepithelial), M2 (extends to lamina propria) and M3 (muscularis mucosa infiltration) [87]. T1b disease is subdivided into Sm1 (involvement of superficial submucosa), Sm2 (extends to center of submucosa) or Sm3 (invades inro deep submucosa) [87]. Disease confined to the mucosa (T1a) has very low probability of locoregional lymph node metastasis, whereas invasion into the submucosa (T1b) significantly raises the changes of lymph node involvement [87]. Metastasis is found in up to 21% of Sm1 lesions and 56% of Sm3 lesions [88,89].

Endoscopic treatment can be in the form of either resection or ablation. One of the benefits of resection is the yield of large specimens that enhance diagnosis and staging [90]. Techniques involving resection include EMR and ESD. EMR is used for T1a lesions, whereas ESD can be used for T1b lesions or large dysplastic lesions (>2 cm) [87]. The risks associated with EMR and ESD include bleeding, and perforation, and the formation of strictures and risks can be higher in ESD [87]. Ablation techniques in the form of argon plasma coagulation, photodynamic therapy, heater probe treatment, cryotherapy and radiofrequency ablation are used for Barrett’s esophagus and intramucosal carcinoma [87]. They are more commonly used in combination with endoscopic resection [87]. Endoscopic treatment for early esophageal malignancy has comparable curative results to surgical resection with the advantage of lower morbidity as compared to surgery [91].

Palliative endoscopic therapy can be utilized for combating the complications of esophageal cancer such as esophageal obstruction and tracheoesophageal fistula and providing enteral nutrition [87]. Treatment includes dilation, debulking, stenting and providing an alternative route for nutritional support [87]. Esophageal dilation is associated with an increased risk for perforation and does provide long term relief as compared to esophageal stenting [92,93]. As the tumor grows, it not only occludes the lumen of the esophagus but also invades the wall of the esophagus and penetrates into the surrounding organs such as the trachea and leads to tracheoesophageal (TE) fistula. A fistula can also develop due to radiation treatment [87]. Self-expandable metal stents (SEMS) are used to relieve dysphagia and allow for per oral nutritional intake and in cases of TE fistula are used to close the communication between the esophagus and trachea and hence to prevent complications such as aspiration [87]. Complications from esophageal stenting include migration, perforation, hemorrhage, tumor growth into the stent, and the formation of fistulas, and when used for TE, fistula can lead to compromise of the airway [87]. The debulking of inoperable tumors can be carried out for obstructing tumors that are not amenable to surgery, and debulking can be achieved with laser ablation, chemicals, or photodynamic therapy (PDT) [87]. PDT is one of the preferred endoscopic techniques for debulking based on efficacy and complications [87]. Lastly, enteral feeding tubes can be placed endoscopically for nutritional support, however they can interfere with surgical methods such as esophagectomy and gastric-pull up [87].

### 5.2. Chemoradiation and Immunotherapy

In addition to staging, determination of HER2 status along with evaluation of microsatellite instability/mismatch repair status and PDL-1 (Programmed death-ligand-1) expression is highly important prior to proceeding with chemotherapy or immunotherapy [94]. Preoperative chemoradiation therapy is linked to favorable outcomes such as increased survival including overall survival and pathologic complete response as compared to surgery or chemotherapy alone [95]. Some of the chemotherapy regimens that are recommended are paclitaxel and carboplatin, FOLFOX (fluorouracil/leucovorin calcium/oxaliplatin), cisplatin and fluorouracil for preoperative chemoradiation.

Perioperative chemotherapy with a regimen consisting of epirubicin, cisplatin, and fluorouracil (ECF) was superior to surgery alone and improved overall and progression-free survival in nonmetastatic stage II and EGF or higher gastric adenocarcinoma [96]. The FLOT trial which compared FLOT (fluorouracil, leucovorin, oxaliplatin, and docetaxel) regimen to the ECF regimen revealed that the FLOT regimen was associated with higher pathologic complete response and overall survival [94,97]. Hence, the favored perioperative regimen is FOLFOX for patients with moderate to good performance status, with ECF being reserved for select cases [94].

Postoperative chemotherapy in addition to chemoradiation has been shown to improve five-year overall survival and recurrence free survival as shown in the Intergroup-0116 trial [94,98]. Definitive chemotherapy can be employed in those unfit for surgery or in unresectable disease [94]. The combined approach with radiation and chemotherapy is associated with improved outcomes including survival as compared to radiation alone [94]. FOLFOX and the combination of cisplatin and fluorouracil are preferred regimens for definitive chemoradiation, although FOLFOX is linked to fewer adverse events [94].

Immunotherapy for advanced and even unresectable disease can be utilized and includes agents such as transtuzumab, ramucirumab, and pemprolizumab [94]. Transtuzumab, a monoclonal antibody which works against human epidermal growth factor receptor 2, is used in HER-2 positive cases of advanced EGJ adenocarcinoma [99]. Ramucirumab, a vascular endothelial growth factor receptor 2 antibody, can be used in patients with advanced or metastatic gastroesophageal cancers that have been previously treated [100]. Pembrolizumab, a monoclonal antibody against PD-1 receptors, is used for inoperable or metastatic solid tumors that have not responded to prior treatment [94].

### 5.3. Surgery

Careful evaluation with a multi-disciplinary team should be performed to select patients that would be fit to undergo surgery. Esophagectomy should be performed for patients that are considered medically suitable for operable esophageal cancer (located >5 c, from the cricopharyngeus) [94]. Definitive chemoradiation should be used for cervical or cervicothoracic esophageal cancers that are located < 5 cm from the cricopharyngeus [94]. For advanced cancers or cases with lymph node metastasis, laparoscopic staging can be performed with peritoneal washings to evaluate for metastasis [94]. During surgery, lymph node dissection has as a goal the removing of at least 15 lymph nodes for evaluation. Prior to proceeding with surgery, nutritional status should be optimized and enteral support may be required, in which case a jejunostomy is preferred over gastrostomy, as gastrostomy can interfere with the formation of the gastric conduit [94]. Overall surgical techniques have made progress from the traditional approaches such as the Ivor-Lewis esophagectomy to less invasive techniques known as minimally invasive esophagectomy using thoracoscopy and/or laparoscopy, such as the mediastinoscopy-assisted transhiatal esophagectomy (MATHE) technique [101]. Minimally invasive techniques may be associated with less risk for complications and shorter hospitalizations [101].

### 5.4. Palliation

Palliative treatment should be offered to those with inoperable, recurrent or metastatic disease, and whether to combine this with systemic therapy is based on the performance status of the patient [94]. Patients with good performance status may benefit from the addition of systemic therapy and it has been linked to improved survival as compared to palliative therapy alone, as shown in a Cochrane systematic review [102]. The review also showed that ramucirumab was linked to improved overall survival and progression free survival, and although there was an increase in adverse events related to treatment, there were no treatment related deaths [94,102]. Quality of life was also noted to be improved as per the report of patients with the addition of systemic therapy to palliative therapy [94]. Dysphagia in particular is quite a distressing symptom and has a great impact on the quality of life and can be treated with a palliative intent irrespective of the performance status [94]. As mentioned above, endoscopic treatments in the form of esophageal stent placement can be carried out. Aside from that, brachytherapy or external beam radiation are other options [94]. Lastly, involvement of the palliative care team early on in the management is crucial as well and not just for advanced disease, but also for early disease so that support and symptomatic management can be achieved.

## 6. Conclusions

Esophageal cancer is a tragic disease that inflicts those from all over the world. The two main histological types are adenocarcinoma and squamous cell carcinoma, which have some distinct characteristics. There are numerous predisposing factors, many of which are modifiable, so patients should be counselled regarding applicable ones. Aside from that the diagnosis is based on history along with physical examination, which provides the basis for clinical suspicion, which gives rise to diagnostic testing including the gold standard test, which is EGD. Once tissue diagnosis is obtained, staging is performed and management can be pursued accordingly. Often the cancer is captured at an advanced stage, hence a multidisciplinary approach with involvement of medical and surgical oncology as well as other specialists such as palliative care early on is crucial. Though the disease has a grave prognosis, there are ongoing efforts in clinical science to broaden the horizon for medical and surgical treatments.

## Figures and Tables

**Figure 1 diseases-10-00044-f001:**
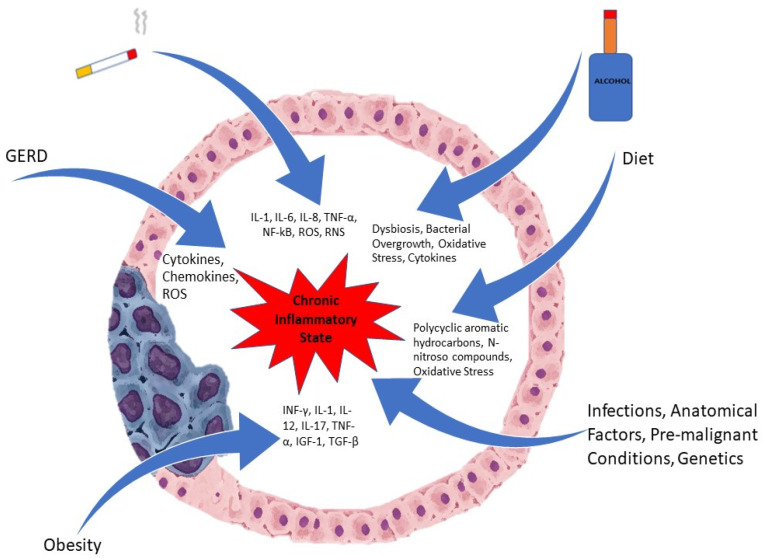
Etiological Factors involved in Esophageal Cancer.

**Figure 2 diseases-10-00044-f002:**
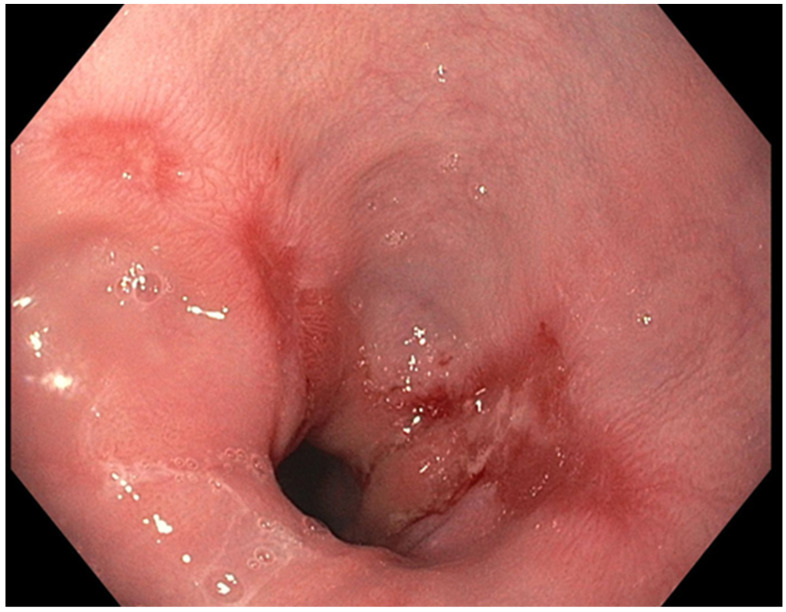
Los Angeles Grade B esophagitis.

**Figure 3 diseases-10-00044-f003:**
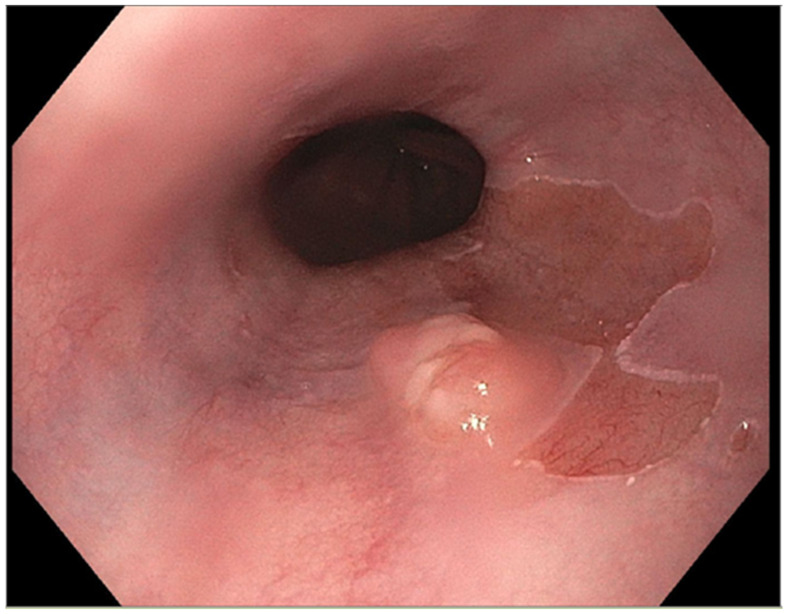
Barrett’s esophagus with nodule.

**Figure 4 diseases-10-00044-f004:**
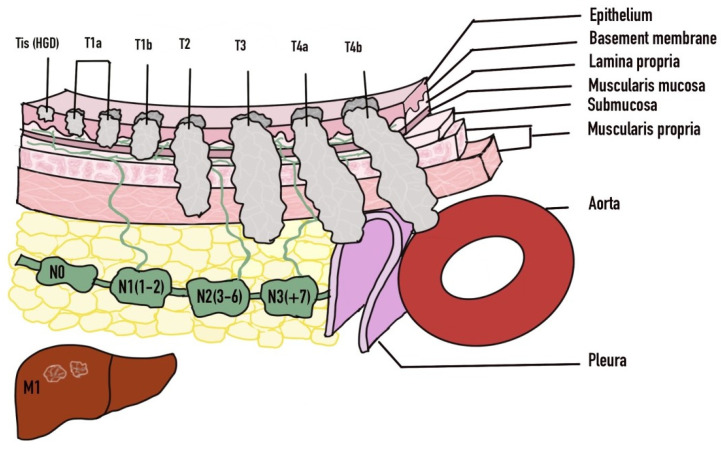
TNM Staging of Esophageal Cancer. Adapted with permission from Ref. [81]. 2016 Luo et al.

**Table 1 diseases-10-00044-t001:** Etiological Factors.

Etiological Factors	
Smoking	Infections
Alcohol	Anatomical factors
Diet	Pre-malignant conditions
Gastroesophageal reflux disease	Genetics
Obesity	

**Table 2 diseases-10-00044-t002:** Diagnostic Modalities.

Diagnostic Modalities	
Barium Esophagogram	Positron Emission Tomography-Computed Tomography Scan
Esophagogastroduodenoscopy	Endoscopic Ultrasound
Multidetector Computed Tomography	

## Data Availability

Not applicable.

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
