# Peer review of "The 8th Wonder of the Cancer World: Esophageal Cancer and Inflammation"

_diseases, 2022, doi:10.3390/diseases10030044_

Round 1
Reviewer 1 Report
The authors have to be congratulated for their thorough and extensive review on esophageal cancer.
It think this paper would benefit from shortening to be aligned with the title. The strength of this paper does not lie in the treatment section, so focussing on inflammation and maybe even cite therapies against this kind of carcinogenesis will imrpove its message.
I didn't find nivolumab as adjuvant treatment option after surgery (NEJM paper)
there are a few typos (line 75, 195, 251,...)
Author Response
Thank you to the reviewer for their kind response and for taking the time to review this paper.
This was intended to be on overall review article that provides some insight into the various components involved in esophageal cancer.
We did mention some immunotherapies with monoclonal antibodies such as transtuzumab, ramucirumab, pemprolizumab.
The typos mentioned in the lines have been edited.
Reviewer 2 Report
1. Abstract needs to be improved in order to include substantial information about the relationship between esophageal cancer and inflammation.
2. As per the journal guidelines “In the text, reference numbers should be placed in square brackets [ ], and placed before the punctuation; for example [1], [1–3] or [1,3].”. Authors have not followed the journal guidelines for reference arrangement.
3. What the authors wishes to convey by this sentence “The incidence has been a drastic change in incidence with a decline in SCC and increase in EAC.” Refer Line no. 36-37, in the Introduction section. Authors should rephrase this sentence to make the meaning clearer.
4. Authors have used an abbreviation “GERD” in Table 1 of 2.1, while its full form was mentioned in 2.1.4.
5. Authors have not expanded “NF-kB” in 2.1.1
6. Authors have abbreviated “reactive oxygen species (ROS)” in Line 62, whereas it should be ideally done in Line 44.
7. Once abbreviated any term, authors should ideally ideally used that abbreviation throughout the manuscript. For instance, after abbreviating ROS in Line 62, authors have still mentioned “reactive oxygen species” in Line 64.
8. Authors should make a section of gut microbiome, inflammation and esophageal cancer.
9. In section 2.1.2, authors have mentioned the linkage of alcohol consumption and cancer. Acetaldehyde was mainly seen as a key factor for it. Usually, acetaldehyde gets easily converted into acetate by enzyme acetaldehyde dehydrogenase, it is even mentioned by authors too. In the rehabilitation centre, drug like disulfiram is being used to treat chronic alcoholism. Disulfiram mainly acts by accumulation of acetaldehyde. Ideally, disulfiram should then be responsible for increase of ESCC. But by referring the article, https://pubmed.ncbi.nlm.nih.gov/29274360/ , it seems like Disulfiram is having anticancer effect on ESCC. This seems contradictory. Authors should discuss it in manuscript and provide a proper reasoning.
10. In Line No. 93 “leads to generation free radicals”, it should be either “leads to generate free radicals” or “leads to generation of free radicals”
11. In Line No. 138, it should be “and can decrease oxidative damage of the DNA and re……”
12. Authors should confirm if Figure 1 and Figure 2 are original or from any other published source?
13. In Line No. 185, it should be “some infectious agents”, not “some infections agents”.
14. Scientific names like “Helicobacter pylori” should be in italics.
15. In section 2.1.9, text is in different font color. Is it for any specific reason, or just a mistake? Similarly, Line no. 396 to 402, are in different format.
16. In Line 251, check “when the disease is As disease progresses,…….”
17. Table 1, Table 2 are not cited in the text.
18. Similarly, Figure 3 is not cited in the text
19. Literature search is very weak. Only 1 article from 2020, 2 from 2021 and none from 2022. Authors must improve it.
Author Response
- Changes made to abstract portion to include more on inflammation.
- References placed in brackets.
- Line re-written.
- GERD elaborated.
- NF-kB expanded
- ROS expanded.
- Changes made for ROS.
- The role of gut microbiome would require extensive addition and would further lengthen the paper.
- We did not include the article about disulfiram in our paper.
- Changes made to sentence.
- Changes made to the sentence.
- Figures 1 and 2 are original
- Changes made.
- Changes made.
- Not able to modify it for some reason.
- Changes made.
- Included tables.
- Included figure
- Given the timeline and nature of the paper, we did our best with the literature search.